# Influence of Slope Direction on the Soil Seed Bank and Seedling Regeneration of *Castanopsis hystrix* Seed Rain

Zong Zhao [1,2], Yong Liu [2,*], Hongyan Jia [1], Wensheng Sun [1], Angang Ming [1], Shengjiang Pang [1], Ning An [1], Jihui Zhang [1], Chuang Tang [1] and Shitao Dong [1]

1    Experimental Center of Tropical Forestry, Chinese Academy of Forestry, Guangxi Youyiguan Forest Ecosystem Research Station, Pingxiang 532600, China; zhaozong@caf.ac.cn (Z.Z.); jiahongyan@caf.ac.cn (H.J.); sunwensheng@caf.ac.cn (W.S.); mingangang@caf.ac.cn (A.M.); pangshengjiang@caf.ac.cn (S.P.); anning@caf.ac.cn (N.A.); zhangjihui@caf.ac.cn (J.Z.); tangchuang@caf.ac.cn (C.T.); dongshitao@caf.ac.cn (S.D.)
2    Key Laboratory for Silviculture and Conservation of Ministry of Education, College of Forestry, Beijing Forestry University, Beijing 100083, China
*    Correspondence: lyong@bjfu.edu.cn; Tel.: +86-10-6233-8994

**Abstract:** Objective: To investigate the impact of different slope directions on the quantity and quality of the soil seed bank and seedling germination process of *Castanopsis hystrix* plantations. Method: Fixed sample plots in forest stands of *Castanopsis hystrix* were established on different slope directions (sunny slope, semi-sunny slope, semi-shady slope, and shady slope). The characteristics of the forest stand were investigated, and per-wood scaling was carried out. The temporal dynamics of the seed rain and seed bank were quantified using seed rain collectors and by collecting soil samples from different depths. The quantity and quality of the seeds were determined, and the vigor of mature seeds was measured throughout the study. Results: (1) The diffusion of *Castanopsis hystrix* seed rain started in mid-September, reached its peak from late October to early November, and ended in mid-December. (2) The dissemination process, occurrence time, and composition of the seed rain varied between the different slope directions. The seed rain intensity on the semi-sunny slope was the highest ($572.75 \pm 9.50$ grains·m$^{-2}$), followed by the sunny slope ($515.60 \pm 10.28$ grains·m$^{-2}$), the semi-shady slope ($382.13 \pm 12.11$ grains·m$^{-2}$), and finally the shady slope ($208.00 \pm 11.35$ grains·m$^{-2}$). The seed rain on the sunny slope diffused earliest and lasted the longest, while the seed rain on the shady slope diffused latest and lasted the shortest time. Seed vigor and the proportion of mature seeds within the seed rain were greatest on the semi-sunny slope, followed by the sunny slope, semi-shady slope, and the shady slope. (3) From the end of the seed rain to August of the following year, the amount of total reserves of the soil seed banks was highest on the semi-sunny slope, followed by the sunny slope then the semi-shady slope, and it was the lowest on the shady slope. The amount of mature, immature, gnawed seeds and seed vigor of the soil seed bank in various slope directions showed a decreasing trend with time. The seeds of the seed bank in all slope directions were mainly distributed in the litter layer, followed by the 0–2 cm humus layer, and only a few seeds were present in the 2–5 cm soil layer. (4) The seedling density of *Castanopsis hystrix* differed significantly on the different slope directions. The semi-sunny slope had the most seedlings, followed by the sunny slope, semi-shady slope, and the shady slope. Conclusions: The environmental conditions of the semi-sunny slope were found to be most suitable for the seed germination and seedling growth of *Castanopsis hystrix*, and more conducive to the regeneration and restoration of its population.

**Keywords:** *Castanopsis hystrix*; slope directions; seed rain; soil seed bank

## 1. Introduction

Seeds and seedlings represent the two most important life-cycle stages during the natural regeneration process of plants, and they have a great significance for the individual reproduction and the restoration of plant populations [1,2]. Seed rain is the process by which seeds spread from the mother tree to the ground by gravity and external forces [3],

and this is the main source of regeneration propagules within forest communities [4]. The temporal and spatial pattern of seed dispersal plays an important role in seed germination and seedling regeneration [5], and it ultimately affects the development and composition of the plant population. The sum of the seeds which are scattered from the mother tree to the litter layer and soil is called the soil seed bank [6]. The soil seed bank, as a potential plant community system, is the result of diffusion of seed rain, and it affects the regeneration ability and direction of the plant population [7]. Seedlings are the main carriers for the regeneration and restoration of plant populations [8], and they are of prime importance for stabilizing the development of forest communities. The quantity and quality of the seed rain, the soil seed bank, and the seedlings directly affect the sexual reproduction and regeneration ability of plant populations. Environmental factors (such as light, temperature, moisture and ground cover) exhibit spatial heterogeneity across forest-land, specifically with different slope directions. This heterogeneity impacts, to different degrees, the quantity and quality of the seed rain, soil seed bank and the seedlings [9]. Therefore, by studying the composition and status of the seed rain, the soil seed bank dynamics, and seedling regeneration on different slope directions, the influence of slope differences on the sexual regeneration ability of plants can be revealed. This knowledge is of great significance for studying plant population renewal dynamics and vegetation restoration [7].

*Castanopsis hystrix* belongs to the Fagaceae family. It is distributed throughout the tropical and subtropical regions of China and has been listed as level II in the "China rare broad-leaved tree species protection" list [10]. *Castanopsis hystrix* not only has a high potential economic value but also has outstanding ecological value in improving soil, water conservation, maintaining regional biodiversity and ecological balance [11,12]. However, in near-mature pure forests, *Castanopsis hystrix* is facing difficulties in regeneration. Coupled with man-made logging and destruction, *Castanopsis hystrix* is experiencing a sharp decline in its population. As a result, the resources of this tree species are on the verge of exhaustion, which is of widespread concern [10]. Previous researches on *Castanopsis hystrix* mainly focused on its ecological and physiological characteristics [12–15], and there are few researches on its the seed rain, seed bank and seedling regeneration. This study aims to clarify the fate of the *Castanopsis hystrix* seeds in the seed rain to seedling germination stages and to reveal the renewal laws of *Castanopsis hystrix* populations. By examining different slope directions, the seed rain and soil seed bank dynamics and seedling regeneration of *Castanopsis hystrix* are studied at the plantation from the Fubo experimental field of the Experimental Center of Tropical Forestry, Chinese Academy of Forestry. The results are expected to provide a scientific basis for improving the management of *Castanopsis hystrix* plantations.

## 2. Materials and Methods

### 2.1. Overview of the Research Area

The research area is located in the Fubo Experimental Field (22°10′ N, 106°50′ E) of the Tropical Forestry Experimental Center (Youyiguan, Guangxi, China), National Orientation Observation and Research Station of Forest Ecosystems of the Chinese Academy of Forestry (CAF). The altitude of this area is 400–550 m and the annual precipitation is 1200–1500 mm. The annual average temperature is 21.9 °C, with a high extreme of 40.0 °C and a low extreme of −2.0 °C. The annual sunshine hours are 1200–1600 h, and this area belongs to the humid and semi-humid monsoon climate zone in the southern subtropics of China. The landform is classified as "low mountain and hills", and the soil is mainly mountain red soil. The soil fertility is considered good. The main arbor species in the forest is *Castanopsis hystrix*. The understory shrub layer includes *Psychotria rubra*, *Antidesma fordii*, and *Embelia laeta*, and the herb layer includes *Lophatherum gracile*, *Dicranopteris linearis*, and *Adiantum flabellulatum*.

### 2.2. Setting of Sample Plots

A pure *Castanopsis hystrix* plantation with a stand age of 34 was examined in this study. This forest was established on the cut-over land of Chinese fir in 1983. In mid-July 2017, *Castanopsis hystrix* stands on four different slope directions (sunny slope, semi-sunny slope, semi-shady slope, and shady slope) were selected. All stands were at a similar altitude and gradient, and they exhibited similar physical conditions. Three sample plots measuring 20 m × 20 m were arranged in each forest stand; thus, a total of 12 sample plots were set up. Within these sample plots, three 5 m × 5 m shrub quadrats and three 1 m × 1 m herb quadrats were arranged along a diagonal line in the upper-left, middle, and lower-right of the plots. Within these quadrats, the height and coverage of shrub and herb plants were measured. Three sampling points were randomly selected in the sample plots of every stand; then the soil samples would be collected in the soil depth of 0–20 cm, 20–40 cm and 40–60 cm, and the soil of the same layer was mixed according to the mass scale in the same plot; then the soil samples were bottled after natural withering and crushing and sieving indoors. Finally, the soil chemical properties were measured. The soil chemical properties of *Castanopsis hystrix* forests are shown in Table 1. The characteristics of the forest stands were investigated, and per-wood scaling was carried out. The stand characteristics of the sample plots are shown in Table 2.

**Table 1.** Nutrient content, C/N ratio and pH value of different soil layers of *Castanopsis hystrix* forests on different slope directions.

| Index | Soil Depth/cm | Sunny Slope | Semi-Sunny Slope | Semi-Shady Slope | Shady Slope |
|---|---|---|---|---|---|
| Organic content/g·kg$^{-1}$ | 0–20 | 66.91 ± 13.30 Aa | 70.48 ± 11.37 Aa | 45.95 ± 4.08 Ab | 47.23 ± 5.13 Ab |
| | 20–40 | 42.32 ± 2.75 Ba | 45.38 ± 7.36 Ba | 38.20 ± 8.82 Aa | 37.12 ± 6.14 Aa |
| | 40–60 | 32.35 ± 5.87 Ba | 33.73 ± 4.72 Ba | 24.22 ± 5.68 Bb | 24.84 ± 4.17 Bb |
| Total nitrogen content/g·kg$^{-1}$ | 0–20 | 1.32 ± 0.11 Aa | 1.96 ± 0.33 Ab | 1.15 ± 0.46 Aa | 1.18 ± 0.77 Aa |
| | 20–40 | 1.02 ± 0.13 Aa | 0.91 ± 0.07 Ba | 0.70 ± 0.03 Bb | 0.66 ± 0.09 Bb |
| | 40–60 | 0.76 ± 0.24 Aa | 0.84 ± 0.12 Ba | 0.69 ± 0.02 Ba | 0.65 ± 0.01 Ba |
| Total phosphorus content/g·kg$^{-1}$ | 0–20 | 0.37 ± 0.03 Aab | 0.42 ± 0.08 Ab | 0.27 ± 0.01 Aa | 0.24 ± 0.05 Aa |
| | 20–40 | 0.26 ± 0.03 Ba | 0.38 ± 0.06 Ab | 0.20 ± 0.02 Ba | 0.19 ± 0.01 Ba |
| | 40–60 | 0.20 ± 0.04 Ca | 0.32 ± 0.03 Ab | 0.15 ± 0.02 Ba | 0.17 ± 0.03 Ba |
| Total potassium content/g·kg$^{-1}$ | 0–20 | 5.68 ± 0.26 Aa | 5.87 ± 0.19 Aa | 3.92 ± 0.32 Ab | 3.80 ± 0.45 Ab |
| | 20–40 | 5.01 ± 0.14 Ba | 5.06 ± 0.25 Bc | 3.46 ± 0.13 ABb | 3.40 ± 0.26 Bb |
| | 40–60 | 4.45 ± 0.07 Ca | 4.52 ± 0.63 Bb | 3.03 ± 0.30 Bb | 3.00 ± 0.51 Bb |
| C/N | 0–20 | 37.90 ± 11.39 Aa | 25.39 ± 5.90 Ab | 19.16 ± 2.56 Ab | 17.25 ± 3.17 Ab |
| | 20–40 | 25.81 ± 4.36 Aa | 28.46 ± 6.37 Aa | 29.80 ± 7.81 Aa | 26.91 ± 4.66 Aa |
| | 40–60 | 27.62 ± 5.75 ABa | 23.22 ± 3.25 Aa | 21.07 ± 6.20 Aa | 20.20 ± 4.18 Aa |
| pH | 0–20 | 4.43 ± 0.09 Aa | 4.98 ± 0.86 Aa | 4.10 ± 0.08 Ab | 4.06 ± 0.09 Ab |
| | 20–40 | 4.65 ± 0.08 Aa | 4.50 ± 0.07 Aa | 4.52 ± 0.12 Ba | 4.46 ± 0.08 Ba |
| | 40–60 | 4.77 ± 0.18 Aa | 4.70 ± 0.15 Aa | 4.59 ± 0.05 Ba | 4.50 ± 0.03 Ba |

The different uppercase letters in the same column and the different lowercase letters in the same row of the same index mean a significant difference at the 0.05 level.

**Table 2.** Stand characteristics of *Castanopsis hystrix* sample plots on slopes of different directions.

| Gradient/° | Slope Position | Slope Direction | Average Height of Trees/m | Average DBH/cm | Depth of Litter Layer/cm | Soil Moisture/% | Canopy Density/% |
|---|---|---|---|---|---|---|---|
| 28–33 | Middle | Sunny slope | 28.0 | 40.6 | 3.8 | 42.2 | 80 |
| 31–35 | Middle | Semi-sunny slope | 27.3 | 38.7 | 4.3 | 45.3 | 83 |
| 29–36 | Middle | Semi-shady slope | 26.8 | 37.0 | 4.6 | 47.5 | 75 |
| 15–20 | Down | Shady slope | 25.1 | 36.2 | 5.0 | 49.7 | 71 |

### 2.3. Investigation of Castanopsis Hystrix Seed Rain

Before setting up the seed rain collectors, the herbaceous plants in the sample plots were removed, and any seeds remaining on the ground were cleared. In mid-July 2017, 5 collection sites were evenly set up along the diagonal of each plot from the lower-left corner to the upper-right corner, and each collection site was equipped with two types of seed rain collectors: (1) the off-ground seed collector, which consists of a 1 m × 1 m PVC pipe collection tube and a 2 mm mesh nylon net; (2) the on-ground seed collector, which was positioned at a distance of 0.5–1.0 m from the off-ground seed collector. To prevent animals from grazing the seeds, four 1 m long PVC pipes were used as supports for the off-ground collector, and the bottom of the net was set at a height of 50–70 cm from the ground. A relatively flat area was chosen for the on-ground seed collector and small shrubs and weeds were removed. The four corners of the nylon net, with an area of 1 m×1 m, were spread flat on the ground and fixed.

Collections using the two types of seed collectors were conducted simultaneously. From the moment the seed rain began until diffusion, the seeds were collected once every 7 days until the end of the seed rain [16]. The quantities of the total number of seeds, mature seeds, immature seeds, insect-eaten seeds, and gnawed seeds were recorded. After each collection, the vigor of all collected mature seeds was identified by the triphenyltetrazolium chloride (TTC) staining method.

### 2.4. Investigation of Castanopsis Hystrix Soil Seed Bank

In December 2017 (the time when the seed rain ended), April 2018 (the time when the seeds began to germinate), June (the time when the seed germination reached its peak), and August (the time before the next seed rain started), the soil seed bank was sampled. A 1 m × 1 m quadrat was randomly set up within a radius of 2.0 m from each on-ground seed collector. A total of 60 small quadrats in the sample plots on the various slope directions were sampled. Soil samples of the litter layer, 0–2 cm humus layer and the 2–5 cm soil layer were collected, bagged and transported back to the laboratory. All seeds, mature seeds, immature seeds, insect-eaten seeds, rotten and gnawed seeds were counted separately. The vigor of the mature seeds was identified by the triphenyltetrazolium chloride (TTC) staining method.

### 2.5. Investigation of 1-Year-Old Castanopsis Hystrix Seedlings

A 1 m × 1 m quadrat was placed near to the on-ground seed collector, within each sample plot. The amount of germinating seeds and surviving seedlings was recorded within this quadrat every two months, from April 2018 (time of seedling emergence) to October (the time no new seedlings appeared).

### 2.6. Data Processing

SPSS software (Version 23.0, Illinois Company, Chicago, IL, USA) was used for the statistical analysis of the data. One-way analysis of variance (one-way ANOVA) was performed to test for differences in seed rain, soil seed bank and 1-year-old seedlings between the slopes of different directions. Duncan's multiple test method was used to determine the significance of the differences ($\alpha = 0.05$). Excel 2010 was used to draw the figures.

## 3. Results

### 3.1. Characteristics of the Castanopsis Hystrix Seed Rain

#### 3.1.1. Temporal Dynamics of Seed Rain Diffusion

As shown in Figure 1, the seed rain of *Castanopsis hystrix* lasted approximately 90 days. The seed rain diffusion could be divided into three stages: the starting period (mid-September), the peak period (from mid-late October to early November), and the ending period (mid-early December). The diffusion of seed rain showed different temporal dynamics during peak periods on the different slope directions. The peak period of seed rain

in sunny and semi-sunny slope directions occurred from 19 October to 9 November, lasting about 22 days. The peak period of seed rain in semi-shady and shady slope directions occurred from 26 October to 9 November, lasting about 15 days. All four slope directions showed obvious seed rain peaks around 26 October.

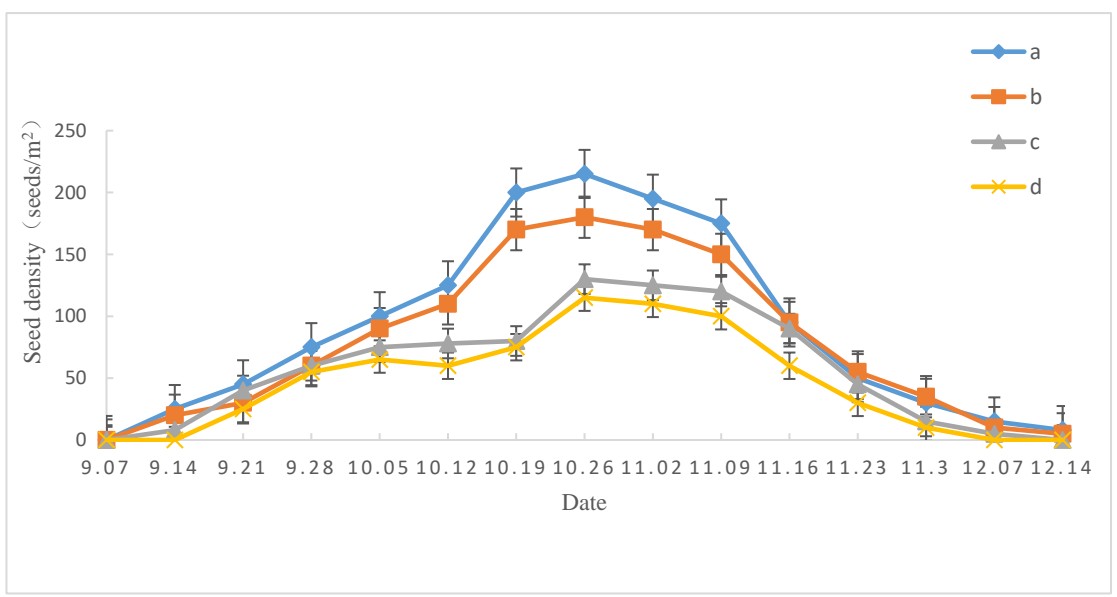

**Figure 1.** Diffusion temporal dynamics of *Castanopsis hystrix* seed rain on different slope directions in 2017. (a) Semi-sunny slope (blue); (b) sunny slope (orange); (c) semi-shady slope (gray); (d) shady slope (yellow).

Figures 1 and 2 display how the seed rain of *Castanopsis hystrix* on all four slope directions was dominated by immature seeds during the starting period of seed diffusion. The proportion of immature seeds to the total amount of seed rain did not significantly differ between the semi-shady slope (28.7%), shady slope (27.9%), semi-sunny slope (27.1%), and the sunny slope (25.3%). During the peak period of seed diffusion, the seed rain on all slope directions was dominated by mature seeds. There was, however, a significant difference in the proportion of mature seeds to the total amount of seed rain on the different slopes (in descending order) was 58.6% on the semi-sunny slope, 57.8% on the sunny slope, 50.3% on the shady slope, and 46.6% on the semi-shady slope. During the ending period of seed diffusion, a small number of mature seeds were dispersed on all slope directions, which did not significantly differ between slopes. In addition, the mature seeds of the seed rain were diffused earlier on the sunny and semi-sunny slopes than on the semi-shady and shady slopes, with the sunny slopes experiencing the earliest mature seed diffusion and shady slopes experiencing the latest.

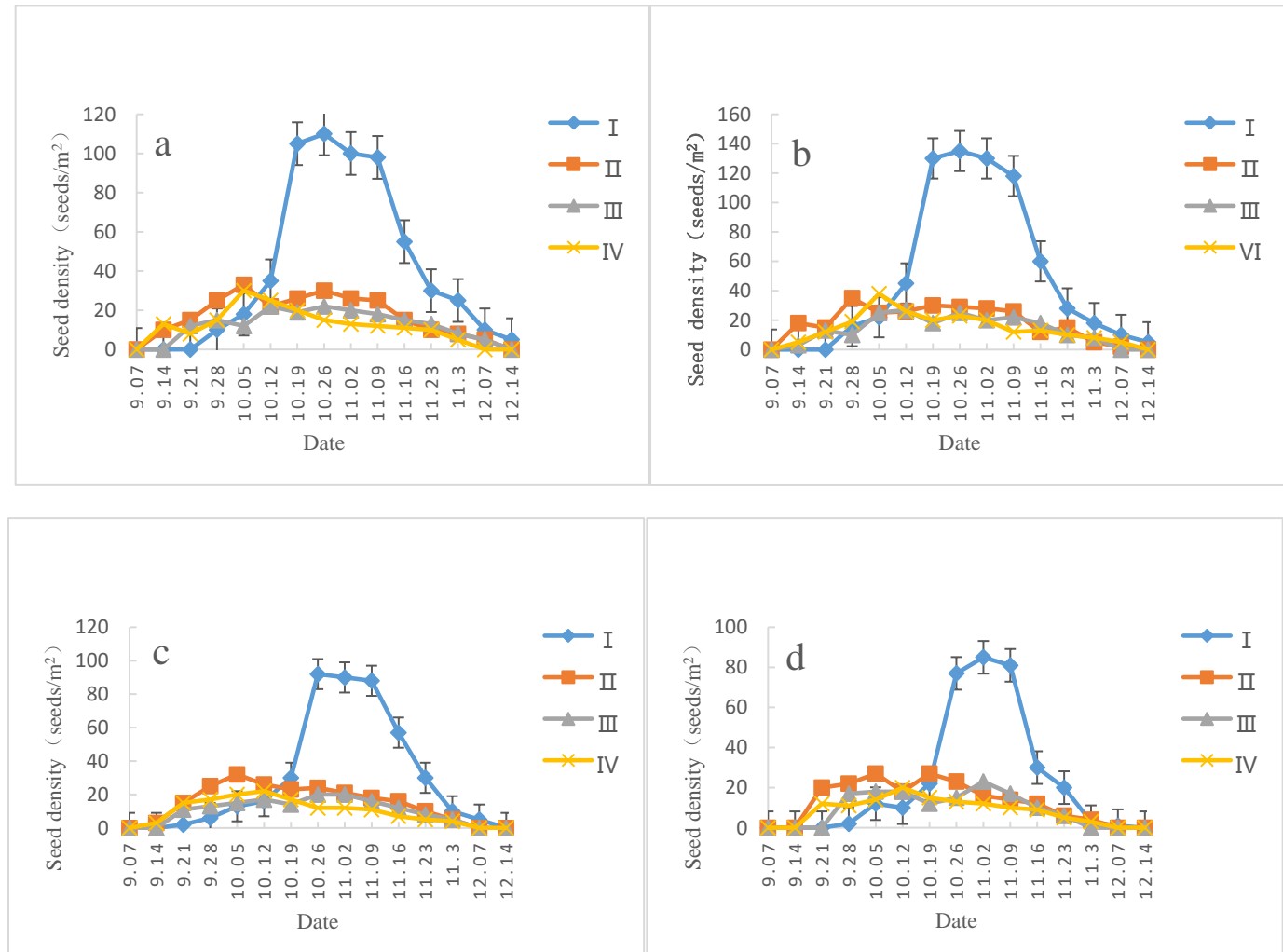

**Figure 2.** (**a**) semi-sunny slope; (**b**) sunny slope; (**c**) semi-shady slope; (**d**) shady slope. Temporal dynamics of *Castanopsis hystrix* seed rain on different slope directions in 2017. (I) Mature seed (blue); (II) immature seed (orange); (III) seed damaged by insects (gray); (IV) nibbled seed (yellow).

### 3.1.2. Composition and Quality of Seed Rain

As shown in Table 3, the total amount of *Castanopsis hystrix* seed rain in the four slope directions was greatest on the semi-sunny slope, followed by the sunny slope, semi-shady slope, and finally the shady slope. The difference in seed rain quantity between the slopes was significant. From the perspective of seed rain composition, the mature seeds on the sunny, semi-sunny, semi-shady, and shady slopes significantly differed and accounted for 52.0%, 55.0%, 48.1%, and 41.6% of the seed rain, respectively. The portion of immature seeds in the total amount of seed rain did not significantly differ between the sunny slope (20.9%), semi-sunny slope (20.0%), semi-shady slope (23.6%), and shady slope (24.0%). The proportion of insect-damaged seeds was 17.9% on the shady slopes, 14.5% on the semi-shady slopes, 13.8% on the sunny slopes, and 12.5% on the semi-sunny slopes, with there being a significant difference between the semi-sunny and shady slopes. The portion of gnawed seeds was greatest on the shady slopes (16.5%), followed by the semi-shady (13.9%), sunny (13.2%), and finally the semi-sunny (12.4%) slopes, with the difference between the semi-sunny slope and the shady slope being significant. There was no significant difference in the seed vigor testing results between the different slopes. The seed vigor of seeds from the semi-sunny slope was the highest (51.0%), followed by the sunny slope (49.6%), semi-shady slope (43.5%), and then the shady slope (34.1%).

**Table 3.** Cumulative characteristics of *Castanopsis hystrix* seed rain on different slope directions.

| Slope Direction | Total Amount of Seed Rain (Seeds·m$^{-2}$) | Amount of Mature Seeds (Seeds·m$^{-2}$) | Amount of Immature Seeds (Seeds·m$^{-2}$) | Amount of Insect-Damaged Seeds (Seeds·m$^{-2}$) | Amount of Gnawed Seeds (Seeds·m$^{-2}$) | Seed Vigor (%) |
|---|---|---|---|---|---|---|
| Sunny slope | 457.80 ± 10.05 b | 238.04 ± 5.48 b | 95.80 ± 3.98 b | 63.40 ± 2.12 a | 60.56 ± 1.84 a | 49.6 ± 3.8 ab |
| Semi-sunny slope | 516.52 ± 10.46 a | 284.32 ± 6.40 a | 103.16 ± 4.20 a | 64.64 ± 2.04 a | 64.40 ± 2.26 a | 51.0 ± 4.2 a |
| Semi-shady slope | 359.00 ± 8.09 c | 172.52 ± 3.72 c | 84.72 ± 3.66 c | 51.92 ± 1.04 b | 49.84 ± 1.72 b | 43.5 ± 3.8 b |
| Shady slope | 310.44 ± 6.31 d | 129.24 ± 2.60 d | 74.44 ± 2.72 d | 55.64 ± 1.50 b | 51.12 ± 1.62 b | 34.1 ± 3.2 c |

The different lowercase letters in the same column mean a significant difference at the 0.05 level.

### 3.2. Characteristics of the Castanopsis Hystrix Soil Seed Bank

3.2.1. Reserves and Dynamics of the Soil Seed Bank

After the diffusion of the seed rain, the quantity of seeds in the soil seed bank being transported and eaten is roughly reflected by the difference between the number of seeds in the on-ground versus the off-ground seed collectors [17]. As shown in Figure 3, the number of seeds within the on-ground seed collectors was more than the number of seeds within the off-ground seed collectors, no matter the slope direction. The proportion of the seeds obtained by the on-ground seed collectors compared to the total seed rain amount was highest on the semi-sunny slope (22.8%), followed by the sunny slope (21.0%), semi-shady slope (18.8%), and the shady slope (15.9%).

Table 4 displays the total seed reserves of the *Castanopsis hystrix* soil seed bank over different periods. The total seed reserves were greatest on the semi-sunny slope, followed by the sunny slope, semi-shady slope, and finally the shady slope. The total seed bank reserves on all slopes showed a decreasing trend with time. The amount of mature, immature, insect-damaged, and gnawed seeds in the seed bank on the various slope directions also showed a decreasing trend over time, while the amount of moldy seeds exhibited an increasing trend. In December 2017 and April 2018, mature seeds made up the majority of the seed bank on all slope directions. The mature seeds on the sunny and semi-sunny slopes accounted for the highest proportion of the total seed bank reserves, which were 52.9% and 48.4%, respectively. In June and August 2018, the amount of moldy and rotten seeds in the soil seed bank on all slope directions increased, accounting for the vast majority of the seed bank. The moldy and rotten seeds of the semi-shady and shady slopes accounted for the highest proportion of total seed bank reserves, which were 51.6% and 75.5%, respectively. The results of mature seed vigor identification showed that for the first three samplings, from December 2017 to August 2018, the seeds on the semi-sunny slope had the highest vigor, followed by the sunny slope, semi-shady slope, and then the shady slope. Afterwards, the seed vigor on all slope directions showed a downward trend. From December 2017 to April 2018, the seed vigor of *Castanopsis hystrix* decreased the fastest, with the decline ratio being highest on the semi-sunny slope (33.8%), followed by the sunny slope (31.2%), semi-shady slope (26.6%), and the shady slope (20.7%). In August 2018, the seeds in the soil seed bank on all slope directions had lost their vitality.

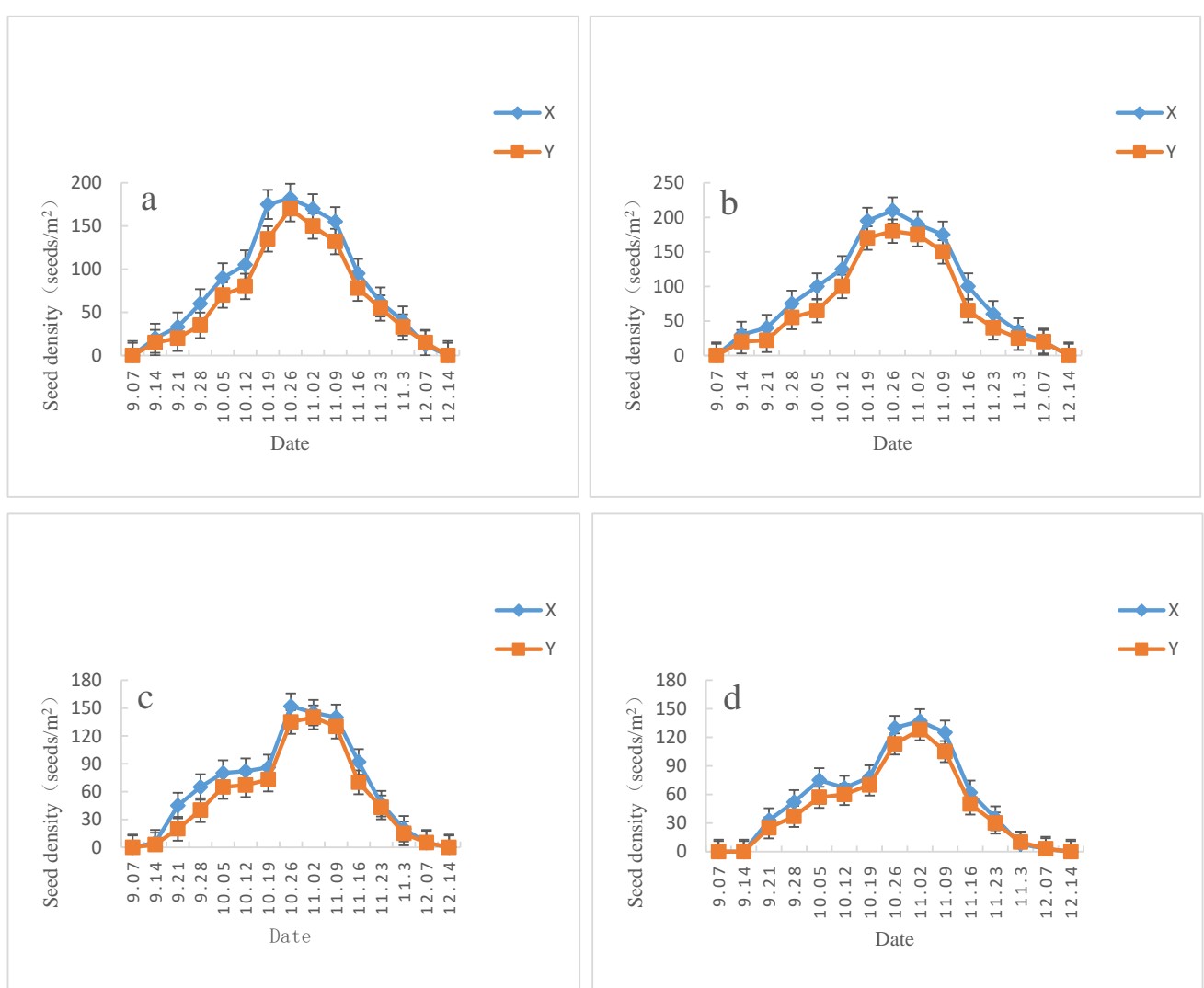

**Figure 3.** (**a**) semi-sunny slope; (**b**) sunny slope; (**c**) semi-shady slope; (**d**) shady slope. *Castanopsis hystrix* seed distribution of two types of seed collectors on different slope directions. (X) On-ground seed collector (blue); (Y) off-ground seed collector (orange).

**Table 4.** Seed vigor and temporal dynamics of the *Castanopsis hystrix* soil seed bank on different slope directions.

| Slope Direction | Date | Seed Bank Density (Seeds·m⁻²) | Amount of Mature Seeds (Seeds·m⁻²) | Vigor (%) | Amount of Immature Seeds (Seeds·m⁻²) | Amount of Insect-Damaged Seeds (Seeds·m⁻²) | Amount of Gnawed Seeds (Seeds·m⁻²) | Amount of Moldy Seeds (Seeds·m⁻²) |
|---|---|---|---|---|---|---|---|---|
| Sunny slope | 2017-12 | 272.80 ± 8.34 a | 136.88 ± 4.62 a | 48.4 ± 2.6 a | 71.16 ± 4.14 a | 28.68 ± 1.36 a | 27.40 ± 1.34 a | 8.68 ± 0.64 d |
| | 2018-04 | 143.64 ± 4.84 b | 62.56 ± 2.50 b | 17.2 ± 1.2 b | 21.28 ± 1.12 b | 18.76 ± 1.14 b | 21.52 ± 1.02 a | 19.52 ± 1.00 c |
| | 2018-06 | 86.28 ± 3.36 c | 15.64 ± 0.88 c | 3.9 ± 0.3 c | 19.24 ± 1.02 b | 16.28 ± 1.04 b | 9.20 ± 0.56 b | 25.92 ± 1.32 b |
| | 2018-08 | 64.20 ± 1.96 c | 5.76 ± 0.34 c | 0.0 ± 0.0 c | 10.56 ± 1.32 c | 11.36 ± 0.70 b | 4.88 ± 0.34 b | 31.64 ± 1.64 a |
| Semi-sunny slope | 2017-12 | 308.68 ± 8.46 a | 163.40 ± 4.40 a | 52.6 ± 2.6 a | 75.04 ± 4.02 a | 29.88 ± 1.48 a | 29.32 ± 1.44 a | 11.04 ± 0.70 d |
| | 2018-04 | 151.24 ± 5.00 b | 73.20 ± 2.72 b | 18.8 ± 1.4 b | 22.52 ± 1.36 b | 21.04 ± 1.10 b | 17.80 ± 1.18 b | 16.68 ± 1.02 c |
| | 2018-06 | 93.68 ± 4.04 c | 18.32 ± 0.98 c | 4.3 ± 0.4 c | 14.88 ± 0.78 c | 18.92 ± 0.96 b | 14.00 ± 0.72 c | 27.56 ± 1.38 b |
| | 2018-08 | 65.76 ± 2.82 d | 7.48 ± 0.20 c | 0.0 ± 0.0 c | 8.44 ± 0.56 d | 8.84 ± 0.48 c | 7.24 ± 0.36 d | 33.76 ± 1.92 a |
| Semi-shady slope | 2017-12 | 239.36 ± 7.18 a | 111.60 ± 4.72 a | 39.3 ± 2.4 a | 72.08 ± 4.10 a | 25.40 ± 1.16 a | 22.52 ± 1.04 a | 7.76 ± 0.54 d |
| | 2018-04 | 87.20 ± 3.78 b | 27.08 ± 1.74 b | 12.7 ± 1.1 b | 15.40 ± 1.00 b | 22.80 ± 1.06 a | 8.88 ± 0.48 b | 13.04 ± 0.68 c |
| | 2018-06 | 61.20 ± 2.66 c | 8.64 ± 0.44 c | 3.3 ± 0.3 c | 6.64 ± 0.54 c | 16.92 ± 1.08 b | 6.32 ± 0.32 b | 22.68 ± 0.98 b |
| | 2018-08 | 49.40 ± 2.08 c | 2.56 ± 0.14 c | 0.0 ± 0.0 c | 5.60 ± 0.46 c | 7.20 ± 0.40 c | 5.08 ± 0.24 b | 28.96 ± 1.56 a |
| Shady slope | 2017-12 | 171.16 ± 5.62 a | 69.28 ± 3.52 a | 31.2 ± 2.2 a | 48.44 ± 1.62 a | 18.20 ± 1.12 a | 26.28 ± 1.24 a | 8.96 ± 0.56 d |
| | 2018-04 | 83.16 ± 3.44 b | 24.56 ± 1.08 b | 10.5 ± 1.0 b | 15.88 ± 0.98 b | 7.72 ± 0.40 b | 15.24 ± 0.66 b | 19.76 ± 0.90 c |
| | 2018-06 | 57.56 ± 2.16 c | 8.92 ± 0.56 c | 2.5 ± 0.2 c | 6.76 ± 0.56 c | 5.92 ± 0.32 b | 6.28 ± 0.52 c | 29.68 ± 1.14 b |
| | 2018-08 | 45.44 ± 2.20 c | 0.64 ± 0.04 d | 0.0 ± 0.0 c | 2.16 ± 0.16 c | 3.48 ± 0.26 b | 4.84 ± 0.34 c | 34.32 ± 1.46 a |

The different lowercase letters in the same column mean a significant difference at the 0.05 level.

### 3.2.2. Vertical Distribution of Soil Seed Bank

Table 5 shows that in terms of vertical spatial distribution, the seeds of the *Castanopsis hystrix* soil seed bank were mainly distributed in the litter layer regardless of the direction of the slope. The overall distribution was greatest in the litter layer, followed by the 0–2 cm humus layer, and subsequently the 2–5 cm soil layer. In the seed banks on different slope directions, the amount of seeds in the litter layer decreased significantly with time, while the amount of seeds in the 0–2 cm humus layer gradually increased with time. Almost no seeds existed in the 2–5 cm soil layer.

**Table 5.** Vertical distribution of *Castanopsis hystrix* soil seed bank on different slope directions (seeds·m$^{-2}$).

| Slope Direction | Date | Litter Layer | 0–2 cm Humus Layer | 2–5 cm Soil Layer |
|---|---|---|---|---|
| Sunny slope | 2017-12 | 133.86 ± 7.60 a | 5.08 ± 0.66 b | 0.0 ± 0.0 c |
| | 2018-04 | 65.38 ± 4.47 b | 12.80 ± 1.26 a | 0.0 ± 0.0 c |
| | 2018-06 | 35.25 ± 3.16 c | 15.18 ± 1.45 a | 0.21 ± 0.02 b |
| | 2018-08 | 23.64 ± 1.73 c | 16.93 ± 1.83 a | 0.88 ± 0.12 a |
| Semi-sunny slope | 2017-12 | 143.60 ± 8.06 a | 9.38 ± 1.12 b | 0.0 ± 0.0 c |
| | 2018-04 | 67.20 ± 4.22 b | 16.80 ± 1.68 a | 0.0 ± 0.0 c |
| | 2018-06 | 37.50 ± 3.70 c | 17.56 ± 1.71 a | 1.25 ± 0.09 b |
| | 2018-08 | 22.18 ± 2.11 c | 19.92 ± 1.92 a | 1.60 ± 0.18 a |
| Semi-shady slope | 2017-12 | 116.93 ± 6.60 a | 7.48 ± 0.42 b | 0.0 ± 0.0 c |
| | 2018-04 | 36.78 ± 3.70 b | 13.88 ± 1.36 a | 0.0 ± 0.0 c |
| | 2018-06 | 22.86 ± 2.46 c | 15.20 ± 1.65 a | 1.05 ± 0.12 b |
| | 2018-08 | 15.66 ± 1.33 c | 16.52 ± 1.84 a | 1.48 ± 0.16 a |
| Shady slope | 2017-12 | 85.36 ± 5.38 a | 6.16 ± 0.68 b | 0.0 ± 0.0 c |
| | 2018-04 | 35.46 ± 3.30 b | 12.49 ± 1.20 a | 0.0 ± 0.0 c |
| | 2018-06 | 21.30 ± 1.83 c | 13.21 ± 1.35 a | 1.80 ± 0.16 b |
| | 2018-08 | 13.82 ± 1.28 c | 15.48 ± 1.57 a | 2.25 ± 0.28 a |

The different lowercase letters in the same column mean a significant difference at the 0.05 level.

### 3.3. Investigation of the Seedlings of Castanopsis Hystrix

By investigating the seedlings of *Castanopsis hystrix* on different slope directions, we found that the seeds began to germinate in early April. The number of regenerated seedlings reached a peak in late June, and no regenerated seedlings appeared in October. It can be seen from Table 6 that the density of surviving seedlings varied significantly on the different slopes. The density of surviving seedlings was the highest on the semi-sunny slope, followed by the sunny slope and the semi-shady slope, while the shady slope had the fewest surviving seedlings.

**Table 6.** Seedling demography of *Castanopsis hystrix* on different slope directions.

| Slope Direction | Density of Regenerated Seedlings (Plants·m$^{-2}$) | Density of Surviving Seedlings (Plants·m$^{-2}$) | Survival Rate (%) |
|---|---|---|---|
| Sunny slope | 29.73 ± 3.31 a | 13.60 ± 1.74 b | 45.75 ± 3.15 b |
| Semi-sunny slope | 37.10 ± 4.55 a | 19.65 ± 2.62 a | 52.96 ± 3.06 a |
| Semi-shady slope | 16.52 ± 2.40 b | 6.80 ± 0.90 bc | 41.16 ± 4.37 b |
| Shady slope | 8.33 ± 1.42 b | 2.37 ± 0.57 c | 28.45 ± 4.58 c |

The different lowercase letters in the same column mean a significant difference at the 0.05 level.

## 4. Discussions

### 4.1. Characteristics of Castanopsis Hystrix Seed Rain

Affected by the forest climate and other environmental factors, the seed rain of many plants belonging to the genus *Castanopsis* (Fagaceae) presents a biennial-bearing phenomenon. The "on" year of seed rain usually appears once every two to three years [18]. The year 2017 was an "on" year of *Castanopsis hystrix* seed rain in this studied area, and the amount of seed rain was relatively large. The temporal dynamics of the seed rain diffusion

could be divided into three stages: starting period, peak period, and ending period [10]. This study found that the seed rain of *Castanopsis hystrix* lasted, in total, around 90 days, while the peak of seeding diffusion occurred from mid-to-late October to early November, lasting 15 days. The peak and duration of the seed diffusion were roughly similar to that of broadleaf trees, such as *Castanopsis sclerophylla* and *Schima superba* [19]. The temporal characteristics of the *Castanopsis hystrix* seed rain diffusion are not only related to its own biological characteristics but also closely related to the habitat conditions [20]. This study found that light is an important factor affecting the maturation of *Castanopsis hystrix* seeds. The light intensity and duration on the sunny and semi-sunny slopes were significantly greater than on the semi-shady and shady slopes. Therefore, the peak duration of the *Castanopsis hystrix* seed rain on the sunny slope was longer than that on the semi-shady and shady slopes, and seed dispersion on the sunny slope occurred the earliest, followed by the semi-sunny, semi-shady, and finally, the shady slope.

The composition of the seed rain during different periods reflects the quality and sexual reproductive ability of the seeds [7]. This study found that in the early stage of the seed rain, the *Castanopsis hystrix* seed rain was mainly composed of immature seeds, insect-damaged seeds, and gnawed seeds. During this early stage, the quantity of immature and gnawed seeds also reached a maximum. At the peak of the seed rain, the number of mature seeds increased rapidly over a relatively short period of time, which is similar to the seed rain distribution pattern of *Quercus liaotungensis* and *Quercus mongolica* [21,22]. The first stage of diffusion occupied part of the predation capacity of the rodents, which reduced the risk of predation on the mature seeds and, thereby, facilitated the spread of the mature seeds and provided seed quality assurance for seedling renewal [23]. This study found that the proportion and vigor of mature seeds on sunny and semi-sunny slopes was greater than on semi-shady and shady slopes. Among them, the semi-sunny slope had the highest proportion and vigor of mature seeds, while the shady slope exhibited the lowest. These results indicate that the slope direction has a significant effect on the composition and vitality of the *Castanopsis hystrix* seed rain. Studies have found that slope direction significantly affects light, temperature, and moisture, which in turn impacts the reproductive regeneration of trees [24]. *Castanopsis hystrix* forests on semi-shady and shady slopes have poorer light and temperature conditions, and higher humidity, which results in a lighter seed weight. Under the influence of external factors, such as wind and rainfall, the seeds fall off easily. Contrastingly, the *Castanopsis hystrix* forests on the sunny and semi-sunny slopes are exposed to good light and temperature conditions and low humidity; thus, the seeds are well developed and do not fall off as easily [25]. Studies have also shown that on slopes with sufficient sunlight, tree seeds are strong and more abundant [26,27]. In this study, although the sunlight conditions on the semi-sunny slope were slightly worse than those on the sunny slope, the temperature, soil moisture, organic matter, and other water and fertilizer conditions were better on the semi-sunny slope. These enhanced soil conditions on the semi-sunny slope resulted in a greater quantity and vigor of *Castanopsis hystrix* seed rain compared to on the sunny slope.

### 4.2. Dynamics of the Castanopsis Hystrix Soil Seed Bank

The soil seed bank is an important part of the life history of plant populations and plays a decisive role in plant population renewal [28,29]. Plant seeds differ according to their biological characteristics and the heterogeneity of their habitats, leading to different seed bank dynamics in different habitats [29]. The temporal dynamics of the soil seed bank reflect the changes in the seed composition of the plant seed bank over different periods [30]. This study showed that directly after the spread of the seed rain of *Castanopsis hystrix* in 2017, the seed density, quantity of mature seeds, and seed vigor on all slopes were at their highest level. In the winter, because the seeds were in a dormant state, the seed bank density and the mature seed vigor on all slopes did not change significantly. In the spring of 2018, the seed bank density and the mature seed vigor on all slope directions greatly reduced due to the increased temperature, germination of some seeds, and greater

predation by animals. In the summer, long-term rainfall resulted in an increased soil moisture and an increase in the activity of soil microorganisms, which greatly increased the proportion of moldy seeds in the seed bank on the various slope directions [31]. Notably, the seed vigor on each slope direction dropped to zero before the seed rain was scattered. Therefore, animal feeding and mildew can be considered the main factors for the reduction of effective seeds in the soil seed bank of *Castanopsis hystrix*. Under natural conditions, the time interval from the beginning of the diffusion of *Castanopsis hystrix* seeds to the complete loss of seed vitality was no more than one year.

The study on the vertical distribution of the *Castanopsis hystrix* soil seed bank showed that because of the large size of *Castanopsis hystrix* seeds, they do not easily pass through the litter layer or the thick isolation zone formed by the root entanglement. Therefore, the seeds in the soil seed bank on all slope directions are mainly concentrated in the litter layer. Under the action of precipitation and their own gravity, coupled with the transportation of seeds by animals, a small amount of seeds do move down and sink into the 0–2 cm humus layer, while very few seeds are present in the 2–5 cm soil layer. Studies have also found that because the seeds of *Castanopsis* are rich in starch, they can attract squirrels and other rodents to carry and hide them, and the vertical distribution of seeds in the soil layer has been changed eventually [32,33]. This study found that the number of seeds obtained by the on-ground seed collectors varied between the different slope directions. This is due to the different species, numbers, and living habits of animals on the different slope directions, resulting in differing amounts and rates of seed ingestion [34]. Studies have found that slope direction has a certain impact on the vegetation coverage of forests [35]. It has also been suggested that in plots with low vegetation coverage, the seeds of *Quercus aliena var. acuteserrata* are eaten by animals more intensively [36]. These previous findings are consistent with the conclusions of this study. The *Castanopsis hystrix* forest standing on shady slopes had the smallest canopy closure and the lowest vegetation coverage, and the seeds can more easily be found by animals. Therefore, the proportion of seeds eaten by animals was the highest. According to the husk shape of the gnawed seeds left in the *Castanopsis hystrix* seed bank, it is speculated that the animals that eat the seeds are rodents.

*4.3. Regeneration of Castanopsis Hystrix Seedlings*

In addition to seed provenance, suitable habitats for seed germination and seedling survival are also important factors affecting plant regeneration [37]. The fate of seeds after the spread of seed rain depends on the screening effect of the habitat. The seeds must fall into a safe habitat to avoid soil poisoning and animal feeding and then must complete the process from seed germination to seedling establishment [2]. Some studies believe that sufficient provenance is one of the basic conditions for forest regeneration [38]. In the spring following the "on" year of *Castanopsis hystrix* seeds, the viable seeds in the soil seed bank can meet the renewal needs of the forest. However, due to the large loss of seeds and the low incidence of seedlings in the current year, it is difficult to achieve the establishment of seedlings [38]. This is consistent with the conclusion of this study. The amount of seed rain in the *Castanopsis hystrix* forest on the semi-sunny slope was large, but the loss of the seed bank was also large, and the seed germination rate and seedling survival rate were low. This study found that the survival rate of *Castanopsis hystrix* seedlings on different slope directions was consistent with the seed rain amount and seed bank density. The seed rain intensity and the density of the seed bank and survival seedlings on the semi-sunny slopes were the highest, while the seed rain intensity and the survival of the seed bank and seedlings on the shady slopes were the lowest.

## 5. Conclusions

There are significant differences in seed rain diffusion, seed bank reserves, and seedling regeneration of *Castanopsis hystrix* between slopes of different directions. The seed rain intensity and the density of the seed bank and survival seedlings on the semi-sunny slopes were the highest, followed by the sunny slope, semi-shady slope, and the shady

slope. Seed vigor and the proportion of mature seeds within the seed rain were greatest on the semi-sunny slope, followed by the sunny slope, semi-shady slope, and the shady slope. Thus, the habitat conditions on semi-sunny slopes are most beneficial to the regeneration of *Castanopsis hystrix*. Therefore, in the management of *Castanopsis hystrix* forests, semi-sunny slopes with superior sunlight and soil conditions should be selected for cultivation first (The soil chemical properties of *Castanopsis hystrix* forests are shown in Table 1). In addition, the soil conditions of the sunny slopes can be improved via the forest management measures. For semi-shady and shady slopes, appropriate intermediate cuttings can be carried out to improve the lighting conditions of the forest. Through clearing the ground cover and using the ecological effects of forest gaps, the sunlight reaching the forest floor would be increased, and the seed loss would be reduced. Therefore, the quality of the soil seed bank and seedlings of *Castanopsis hystrix* could be improved and could then promote the natural regeneration of *Castanopsis hystrix* and the sustainable development of its population.

**Author Contributions:** Conceptualization, Y.L., H.J. and W.S. Methodology, Y.L., H.J. and Z.Z. Field measurement, Z.Z., A.M., S.P., N.A., J.Z., C.T. and S.D. Statistical processing, Z.Z. Writing and editing, Z.Z. and Y.L. All people listed as authors approved the final submitted version and agreed to share collective responsibility and accountability for the work. All authors have read and agreed to the published version of the manuscript.

**Funding:** The work was supported by the basic scientific research project of Chinese Academy of Forestry (CAFYBB2020ZA001-3).

**Conflicts of Interest:** The authors declare no conflict of interest.

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
