# Peer review of "Influence of Slope Direction on the Soil Seed Bank and Seedling Regeneration of Castanopsis hystrix Seed Rain"

_forests, doi:10.3390/f12040500_

Round 1

Reviewer 1 Report

More references should be cited in the introduction and discussion. The discussion should be enriched with new references and the number of references should be added because 29. is not enough for this type of research. Materials and methods of work are very poor, it is not described how the vitality of seeds was assessed (ISTA or some other rules). The tables from the working methods are your results and you should show them in the results section. The images from the results chapter can be presented more clearly graphically. You are missing conclusions and this should be added in the paper. There are a lot of mistakes with regard to spaces, correct citation of literature as stated by the journal Forests and professional terminology in non-English, which is quite poorly translated for you.

Reviewer 2 Report

In general the quality of the paper is good, please see the attached file for more specific comments. 

Round 2

Reviewer 1 Report

The integrity of the work must be observed in the future .

This manuscript is a resubmission of an earlier submission. The following is a list of the peer review reports and author responses from that submission.